# HIGH-DIMENSIONAL BAYESIAN OPTIMIZATION WITH GROUP TESTING

## ABSTRACT

Bayesian optimization is an effective method for optimizing expensive-to-evaluate black-box functions. High-dimensional problems are particularly challenging as the surrogate model of the objective suffers from the curse of dimensionality, which makes accurate modeling difficult. We propose a group testing approach to identify active variables to facilitate efficient optimization in these domains. The proposed algorithm, Group Testing Bayesian Optimization (`GTBO`), first runs a testing phase where groups of variables are systematically selected and tested on whether they influence the objective. To that end, we extend the well-established group testing theory to functions of continuous ranges. In the second phase, `GTBO` guides optimization by placing more importance on the active dimensions. By exploiting the axis-aligned subspace assumption, `GTBO` is competitive against state-of-the-art methods on several synthetic and real-world high-dimensional optimization tasks. Furthermore, `GTBO` aids in the discovery of active parameters in applications, thereby enhancing practitioners' understanding of the problem at hand.

## 1 INTRODUCTION

Noisy and expensive-to-evaluate black-box functions occur in many practical optimization tasks, including material design (Zhang et al., 2020), hardware design (Nardi et al., 2019; Ejjeh et al., 2022), hyperparameter tuning (Kandasamy et al., 2018; Ru et al., 2020; Chen et al., 2018), and robotics (Calandra et al., 2014; Berkenkamp et al., 2021; Mayr et al., 2022). Bayesian Optimization (BO) is an established framework that allows the optimization of such problems in a sample-efficient manner (Shahriari et al., 2016; Frazier, 2018). While BO is a popular approach for black-box optimization problems, its susceptibility to the curse of dimensionality has impaired its applicability to high-dimensional problems, such as in robotics (Calandra et al., 2014), joint neural architecture search and hyperparameter optimization (Bansal et al., 2022), drug discovery (Negoescu et al., 2011), chemical engineering (Burger et al., 2020), and vehicle design (Jones, 2008).

In recent years, efficient approaches have been proposed to tackle the limitations of BO in high dimensions. Many of these approaches assume the existence of a low-dimensional *active subspace* of the input domain that has a significantly larger impact on the optimization objective than its complement (Wang et al., 2016; Letham et al., 2020). In many applications, the active subspace is further assumed to be axis-aligned (Nayebi et al., 2019; Eriksson & Jankowiak, 2021; Papenmeier et al., 2022; 2023), i.e., only a set of all considered variables impact the objective. The assumption of axis-aligned subspaces is foundational for many successful approaches (Nayebi et al., 2019; Eriksson & Jankowiak, 2021; Song et al., 2022). While it is assumed to hold true for many practical problems, such as in engineering or hyperparameter tuning, few approaches use it to its full extent: either they rely on random embeddings that need to be higher-dimensional than the active subspace to have a high probability of containing the optimum (Nayebi et al., 2019; Papenmeier et al., 2022), or they carry out the optimization in the full-dimensional space while disregarding some of the dimensions dynamically (Eriksson & Jankowiak, 2021; Hvarfner et al., 2023; Song et al., 2022).

Knowing the active dimensions of a problem yields additional insight into the application, allowing the user to decide which problem parameters deserve more attention in the future. When the active subspace is axis-aligned, finding the active dimensions can be framed as a feature selection problem. A straightforward approach is first to learn the active dimensions using a dedicated feature selection approach and subsequently optimize over the learned subspace. We propose to initially find the active

dimensions using an information-theoretic approach built around the well-established theory of group testing (Dorfman, 1943). Group testing is the problem of finding several active elements within a larger set by iteratively testing groups of elements to distinguish active members. We develop the theory needed to transition noisy group testing, which otherwise only allows binary observations, to work with evaluations of continuous black-box functions. This enables group testing in BO and other applications, such as feature selection for regression problems. The contributions of this work are:

1. We extend the theory of group testing to feature importance analysis in a continuous setting tailored towards Gaussian process modeling.
2. We introduce Group Testing Bayesian Optimization (GTBO), a novel BO method that finds active variables by testing groups of variables using a mutual information criterion.
3. We demonstrate that GTBO frequently outperforms state-of-the-art high-dimensional methods and reliably identifies active dimensions with high probability when the underlying assumptions hold.

## 2 BACKGROUND

### 2.1 HIGH-DIMENSIONAL BAYESIAN OPTIMIZATION

We aim to find a minimizer $\boldsymbol{x}^* \in \arg\min_{\boldsymbol{x} \in \mathcal{X}} f(\boldsymbol{x})$ of the black-box function $f(\boldsymbol{x}) : \mathcal{X} \to \mathbb{R}$, over the $D$-dimensional input space $\mathcal{X} = [0, 1]^D$. We assume that $f$ can only be observed point-wise and that the observation is perturbed by noise, $y(\boldsymbol{x}) = f(\boldsymbol{x}) + \varepsilon_i$ with $\varepsilon_i \sim \mathcal{N}(0, \sigma_n^2)$, where $\sigma_n^2$ is the *noise variance*. We further assume $f$ to be expensive to evaluate, so the number of function evaluations is limited. In this work, we consider problems of high dimensionality $D$, where only $d_e$ dimensions are *active*, and the other $D - d_e$ dimensions are *inactive*. Here, inactive means that the function value changes only marginally along the inactive dimensions compared to the active dimensions to the extent that satisfactory optimization performance can be obtained by considering the active dimensions alone. The assumption of active and inactive dimensions is equivalent to assuming an *axis-aligned* active subspace (Eriksson & Jankowiak, 2021), i.e., a subspace that can be obtained by removing the inactive dimensions. We refer the reader to Frazier (2018) for an in-depth introduction to BO.

**Low-dimensional active subspaces and linear embeddings.** Using linear embeddings is a common approach when optimizing high-dimensional functions that contain a low-dimensional active subspace. REMBO (Wang et al., 2016) shows that a random embedded subspace with at least the same dimensionality as the active subspace is guaranteed to contain an optimum if the subspace is unbounded. This inspired the idea to run BO in embedded subspaces. Alebo (Letham et al., 2018) presents a remedy to shortcomings in the search space design of REMBO. In particular, bounds from the original space are projected into the embedded space, and the kernel in the embedded space is adjusted to preserve distances from the original space. Other approaches, such as HeSBO (Nayebi et al., 2019), and BAxUS (Papenmeier et al., 2022), assume the embedded space to be axis-aligned and propose a projection based on the count-sketch algorithm where each dimension in the original space is assigned to exactly one dimension in the embedded space. Bounce (Papenmeier et al., 2023) extends those approaches to combinatorial and mixed spaces with an embedding that allows mixing of various variable types.

**High-dimensional Bayesian optimization in the input space.** Another approach that assumes an axis-aligned active subspace is SAASBO (Eriksson & Jankowiak, 2021), which adds a strong sparsity-inducing prior to the hyperparameters of the Gaussian process model. This makes SAASBO prioritize fewer active dimensions unless the data strongly suggests otherwise. It employs a fully Bayesian treatment of the model hyperparameters. The cubic scaling of the inference procedure lends SAASBO impractical to run for more than a few hundred samples. Another popular approach is to use trust regions (Pedrielli & Ng, 2016; Regis, 2016). Instead of reducing the number of dimensions, TuRBO (Eriksson et al., 2019) optimizes over a hyper-rectangle in input space. This makes the algorithm more local to counteract the over-exploration exhibited by traditional BO in high dimensions. Even though TuRBO operates in the full input dimensionality and might not scale to arbitrarily high-dimensional problems, it has shown remarkable performance in several applications.

**Active subspace learning.** In this paper, we resolve to a more direct approach, where we learn the active subspace explicitly. This is frequently denoted by *active subspace learning*. A common approach is to divide the optimization into two phases. The first phase involves selecting points and analyzing the structure to find the subspace. An optimization phase then follows on the subspace that was identified. The initial phase can also be used alone to gain insights into the problem. One of the more straightforward approaches is to look for linear trends using methods such as *Principal Component Analysis* (PCA Ulmasov et al. (2016)) or *Partial Least Squares* (PLS Bouhlel et al. (2016)). Djolonga et al. (2013) use low-rank matrix recovery with directional derivatives with finite differences to find the active subspace. If gradients are available, the active subspace is spanned by the eigenvectors of the matrix $C := \int_\chi \nabla f(x)(\nabla f(x))^T dx$ with non-zero eigenvalues. This is used by Constantine et al. (2015) and Wycoff et al. (2021) to show that $C$ can be estimated in closed form for GP regression. Large parts of the active subspace learning literature yield non-axis-aligned subspaces. This can be more flexible in certain applications but often provides less intuition about the problem. We refer to the survey by Binois & Wycoff (2022) for a more in-depth introduction to active subspace learning.

## 2.2 GROUP TESTING

Group testing (Aldridge et al., 2019) is a methodology for identifying elements with some low-probability characteristic of interest by jointly evaluating groups of elements. It was initially developed to test for infectious diseases in larger populations but has later been applied in quality control (Cuturi et al., 2020), communications (Wolf, 1985), molecular biology (Balding et al., 1996; Ngo & Du, 2000), pattern matching (Macula & Popyack, 2004; Clifford et al., 2010), and machine learning (Zhou et al., 2014).

Group testing can be subdivided into two paradigms: *adaptive* and *non-adaptive*. In adaptive group testing, tests are conducted sequentially, and previous results can influence the selection of subsequent groups, whereas, in the non-adaptive setting, the complete testing strategy is provided up-front. A second distinction is whether test results are perturbed by evaluation noise. In the noisy setting, there is a risk that testing a group with active elements would show a negative outcome and vice versa. Our method presented in Section 3 can be considered an adaptation of noisy adaptive group testing (Scarlett, 2018).

Cuturi et al. (2020) present a *Bayesian Sequential Experimental Design* approach for binary outcomes, which at each iteration selects groups that maximize one of two criteria: the first one is the mutual information between the elements' probability of being active, $\boldsymbol{\xi}$, in the selected group and the observation. The second is the area under the marginal encoder's curve (AUC). As the distribution over the active group $p(\boldsymbol{\xi})$ is a $2^n$-dimensional vector, it quickly becomes impractical to store and update. Consequently, they propose using a *Sequential Monte Carlo* (SMC) sampler (Del Moral et al., 2006), representing the posterior probabilities by a number of weighted particles.

## 3 GROUP TESTING FOR BAYESIAN OPTIMIZATION

Our proposed method, GTBO, fully leverages the assumption of axis-aligned active subspaces by explicitly identifying the active dimensions. This section describes how we adapt the group testing methodology to find active dimensions in as few evaluations as possible. Subsequently, we use the information to set strong priors for the GP length scales, providing the surrogate model with the knowledge about which features are active.

**Noisy adaptive group testing.** The underlying assumption is that a population of $n$ elements exists, each of which either possesses or lacks a specific characteristic. We refer to the subset of elements with this characteristic as the active group, considering the elements belonging to this group as active. We let the random variable $\xi_i$ denote whether the element $i$ is active ($\xi_i = 1$), or inactive ($\xi_i = 0$), similar to Cuturi et al. (2020) who studied binary outcomes. The state of the whole population can be written as the random vector $\boldsymbol{\xi} = \{\xi_1, \ldots, \xi_n\} \in \{0, 1\}^n$. Further, we denote the true state as $\boldsymbol{\xi}^*$.

We aim to uncover each element's activeness by performing repeated group tests. We write $\boldsymbol{g}$ as a binary vector $\boldsymbol{g} = \{g_1, \ldots, g_n\} \in \{0, 1\}^n$, such that $g_i = 1$ signifies that element $i$ belongs to the group. In noisy group testing, the outcome of testing a group is a random event described by the

random variable $Y(\boldsymbol{g}, \boldsymbol{\xi}^*) \in \{0,1\}$. A common assumption is that the probability distribution of $Y(\boldsymbol{g}, \boldsymbol{\xi}^*)$ only depends on whether group $\boldsymbol{g}$ contains any active elements, i.e., $\boldsymbol{g}^\intercal \boldsymbol{\xi}^* \geq 1$. In this case, one can define the sensitivity $P(Y(\boldsymbol{g}, \boldsymbol{\xi}^*) = 1 \mid \boldsymbol{g}^\intercal \boldsymbol{\xi}^* \geq 1)$ and specificity $P(Y(\boldsymbol{g}, \boldsymbol{\xi}^*) = 0 \mid \boldsymbol{g}^\intercal \boldsymbol{\xi}^* = 0)$ of the test setup.

As we assume the black-box function $f$ to be expensive to evaluate, we select groups $\boldsymbol{g}_t$ to learn as much as possible about the distribution $\boldsymbol{\xi}$ for a limited number of iterations $t = 1 \dots T$. In other words, we want the posterior probability mass function over $\boldsymbol{\xi}$: $P\left[\boldsymbol{\xi} \mid Y(\boldsymbol{g}_1) = y_1, \dots, Y(\boldsymbol{g}_T) = y_T\right]$, to be as informative as possible.

We can identify the active variables by modifying only a few variables in the search space and observing how the objective changes. Intuitively, if the function value remains approximately constant after perturbing a subset of variables from the default configuration, it suggests that these variables are inactive. On the contrary, if a specific dimension $i$ is included in multiple subsets and the output changes significantly upon perturbation of those subsets, this suggests that dimension $i$ is highly likely to be active.

Unlike in the traditional group testing problem, where outcomes are binary, we work with continuous, real-valued function observations. To evaluate how a group of variables affects the objective function, we first evaluate a *default* configuration in the center of the search space, $\boldsymbol{x}_{\text{def}}$, and then vary the variables in the group and study the difference. We use the group notation $\boldsymbol{g}_t \in \{0,1\}^D$ as a binary indicator denoting which variables we change in iteration $t$. Similarly, we reuse the notation that the random variable $\boldsymbol{\xi}$ denotes the active dimensions, and the true state is denoted by $\boldsymbol{\xi}^*$.

The new configuration to evaluate is selected as

$$\boldsymbol{x}_t = \boldsymbol{x}_{\text{def}} \oplus (\boldsymbol{g}_t \otimes \boldsymbol{U}), \tag{1}$$

where $\boldsymbol{U}_{i,j} \sim \mathcal{U}(-0.5, 0.5)$, $\oplus$ is element-wise addition, and $\otimes$ is element-wise multiplication. Note that a point $\boldsymbol{x}_t$ is always associated with a group $\boldsymbol{g}_t$ that determines along which dimensions $\boldsymbol{x}_t$ differs from the default configuration. For the newly obtained configuration $\boldsymbol{x}_t$, we must assess whether $|f(\boldsymbol{x_t}) - f(\boldsymbol{x}_{\text{def}})| \gg 0$, which would indicate that the group $\boldsymbol{g}_t$ contains active dimensions, i.e., $\boldsymbol{g}_t^\intercal \boldsymbol{\xi}^* \geq 1$. However, as we generally do not have access to the true values $f(\boldsymbol{x}_{\text{def}})$ or $f(\boldsymbol{x}_t)$ due to observation noise, we use an estimate $\hat{f}(\boldsymbol{x})$.

Since $f$ can only be observed with Gaussian noise of unknown variance $\sigma_n^2$, there is always a non-zero probability that a high difference in function value occurs between $\boldsymbol{x}$ and $\boldsymbol{x}_{\text{def}}$ even if group $\boldsymbol{g}$ contains no active dimensions. Therefore, we take a probabilistic approach, which relies on two key assumptions:

1. $Z_t := \hat{f}(\boldsymbol{x}_t) - \hat{f}(\boldsymbol{x}_{\text{def}}) \sim \mathcal{N}(0, \sigma_n^2)$ if $\boldsymbol{g}_t^\intercal \boldsymbol{\xi} = 0$, i.e., function values follow the noise distribution if the group $\boldsymbol{g}_t$ contains no active dimensions.
2. $Z_t := \hat{f}(\boldsymbol{x}_t) - \hat{f}(\boldsymbol{x}_{\text{def}}) \sim \mathcal{N}(0, \sigma^2)$ if $\boldsymbol{g}_t^\intercal \boldsymbol{\xi} \geq 1$, i.e., function values are drawn from a zero-mean Gaussian distribution with the function-value variance if the group $\boldsymbol{g}_t$ contains active dimensions.

The first assumption follows from the assumption of Gaussian observation noise and an axis-aligned active subspace. The second assumption follows from a GP prior assumption on $f$, under which $\hat{f}(x_t)$ is normally distributed. As we are only interested in the change from $f(\boldsymbol{x}_{\text{def}})$, we assume this distribution to have mean zero.

We estimate the noise variance, $\sigma_n^2$, and function-value variance, $\sigma^2$, based on an assumption on the maximum number of active variables. First we evaluate $f$ at the default configuration $\boldsymbol{x}_{\text{def}}$. We then split the dimensions into several roughly equally sized bins. For each bin, we evaluate $f$ on the default configuration perturbed along the direction of all variables in that bin and compare the result with the default value. We then estimate the function variance as the empirical variance among the `max_act` largest such differences, and the noise variance as the empirical variance among the rest. Here, `max_act` represents the assumed maximum number of active dimensions. If the assumption holds, there can be no active dimensions in the noise estimate, which is more sensitive to outliers. It is important that it is indeed an upper bound, as the method is more sensitive to estimating the noise from active dimensions than vice versa. An example of this is shown in Appendix D.

Under Assumptions 1 and 2, the distribution of $Z_t$ depends only on whether $\boldsymbol{g}_t$ contains active variables. Given the probability distribution over population states $p(\boldsymbol{\xi})$, the probability that $\boldsymbol{g}_t$

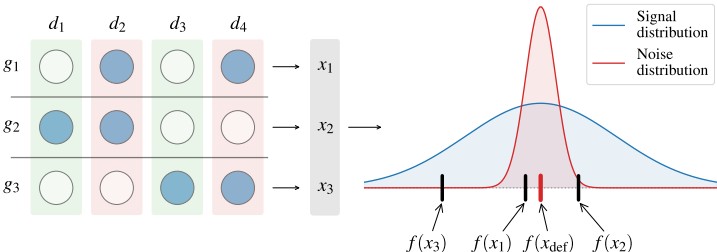

Figure 1: GTBO assumes an axis-aligned subspace. A point $x_1$ that only varies along inactive dimensions ($d_2$ and $d_4$) obtains a similar function value as the default configuration ($x_{\text{def}}$). Points $x_2$ and $x_3$ that vary along active dimensions ($d_1$ and $d_3$) have a higher likelihood under the signal distribution than under the noise distribution.

contains any active elements is

$$p(\boldsymbol{g}_t^\mathsf{T} \boldsymbol{\xi}^* \geq 1) = \sum_{\boldsymbol{\xi} \in \{0,1\}^D} \delta_{\boldsymbol{g}_t^\mathsf{T} \boldsymbol{\xi} \geq 1} p(\boldsymbol{\xi}). \tag{2}$$

We exemplify this in Figure 1. Here, three groups are tested sequentially, out of which the second and third contain active variables. The three corresponding configurations, $x_1$, $x_2$, and $x_3$, give three function values shown on the right-hand side. As observing $f(x_1)$ is more likely under the noise distribution, $g_1$ has a higher probability of being inactive. Similarly, as $f(x_2)$ and $f(\hat{x}_3)$ are more likely to be observed under the signal distribution, $g_1$ and $g_2$ are more likely to be active.

**Estimating the group activeness probability.** Equation (2) requires summing over $2^D$ possible activity states, which, for higher-dimensional functions, becomes prohibitively expensive. Instead, we use an SMC sampler with $M$ particles $\{\boldsymbol{\xi}_1, \ldots, \boldsymbol{\xi}_M\}$ and particle weights $\{\omega_1, \ldots, \omega_M\}$. Each particle $\boldsymbol{\xi}_k \in \{0,1\}^D$ represents a possible ground truth. We follow the approach presented in Cuturi et al. (2020) and use a modified Gibbs kernel for discrete spaces (Liu, 1996). We then estimate the probability $p(\boldsymbol{g}_t^\mathsf{T} \boldsymbol{\xi}^* \geq 1)$ of a group $\boldsymbol{g}_t$ to be active by

$$\hat{p}(\boldsymbol{g}_t^\mathsf{T} \boldsymbol{\xi}^* \geq 1) = \sum_{k=1}^{M} \omega_k \delta_{\boldsymbol{g}_t^\mathsf{T} \boldsymbol{\xi}_k \geq 1}. \tag{3}$$

**Choice of new groups.** We choose new groups to maximize the information obtained about $\boldsymbol{\xi}$ when observing $Z_t$. This can be achieved by maximizing their *mutual information* (MI). Under Assumptions 1 and 2, we can write the MI as

$$I(\boldsymbol{\xi}, Z_t) = H(Z_t) - H(Z_t|\boldsymbol{\xi}) = H(Z_t) - \sum_{\xi \in \{0,1\}^D} p(\xi) H(Z_t|\boldsymbol{\xi} = \xi) \tag{4}$$

$$= H(Z_t) - [p(\boldsymbol{g}_t^\mathsf{T} \boldsymbol{\xi}^* \geq 1) H(Z_t|\boldsymbol{g}_t^\mathsf{T} \boldsymbol{\xi} \geq 1) + p(\boldsymbol{g}_t^\mathsf{T} \boldsymbol{\xi}^* = 0) H(Z_t|\boldsymbol{g}_t^\mathsf{T} \boldsymbol{\xi} = 0)]$$

$$= H(Z_t) - \frac{1}{2}[p(\boldsymbol{g}_t^\mathsf{T} \boldsymbol{\xi}^* = 0) \log(2\sigma_n^2 \pi e) + p(\boldsymbol{g}_t^\mathsf{T} \boldsymbol{\xi}^* \geq 1) \log(2\sigma^2 \pi e)].$$

Since $Z_t$ is modeled as a Gaussian Mixture Model (GMM), its entropy $H(Z_t)$ has no known closed-form expression (Huber et al., 2008), but can be approximated using Monte Carlo:

$$H(Z_t) = \mathbb{E}[-\log Z_t] \approx -\frac{1}{N} \sum_{i=1}^{N} \log p(z_t^i), \tag{5}$$

and $z_t^i \sim \mathcal{N}(0, \sigma^2)$ with probability $\hat{p}(\boldsymbol{g}_t^\mathsf{T} \boldsymbol{\xi}^* \geq 1)$ and $z_t^i \sim \mathcal{N}(0, \sigma_n^2)$ with probability $\hat{p}(\boldsymbol{g}_t^\mathsf{T} \boldsymbol{\xi}^* = 0)$.

**Maximizing the mutual information.** GTBO optimizes the MI using a multi-start forward-backward algorithm (Russell, 2010). First, several initial groups are generated by sampling from the

prior and the posterior over $\boldsymbol{\xi}$. Then, elements are greedily added for each group in a *forward phase* and removed in a subsequent *backward phase*. In the forward phase, we incrementally include the element that results in the greatest MI increase. Conversely, in the backward phase, we eliminate the element that contributes the most to MI increase. Each phase is continued until no further elements are added or removed from the group or a maximum group size is reached. We limit the maximum group size because a large group with many variables of slight impact on the function value could, taken together, make the group appear "active". Finally, the group with the largest MI is returned.

**Updating the activeness probability.** Once we have selected a new group $\boldsymbol{g}_t$ and observed the corresponding function value $z_t$, we update our estimate of $\hat{p}(\boldsymbol{\xi}_k)$ for each particle $k$:

$$\hat{p}^t(\boldsymbol{\xi}_k) \propto \hat{p}^{t-1}(\boldsymbol{\xi}_k) p(z_t|\boldsymbol{\xi}_k) \tag{6}$$

$$\propto \hat{p}^{t-1}(\boldsymbol{\xi}_k) \begin{cases} p(z_t|\boldsymbol{g}_t^\mathsf{T}\boldsymbol{\xi}_k \geq 1) & \text{if } \boldsymbol{g}_t^\mathsf{T}\boldsymbol{\xi}_k \geq 1 \\ p(z_t|\boldsymbol{g}_t^\mathsf{T}\boldsymbol{\xi}_k = 0) & \text{if } \boldsymbol{g}_t^\mathsf{T}\boldsymbol{\xi}_k = 0, \end{cases} \tag{7}$$

where $p(z_t|\boldsymbol{g}_t^\mathsf{T}\boldsymbol{\xi}_k = 0)$ and $p(z_t|\boldsymbol{g}_t^\mathsf{T}\boldsymbol{\xi}_k = 1)$ are Gaussian likelihoods.

Assuming that the probabilities of dimensions to be active are independent, the prior probability is given by $\hat{p}^0(\boldsymbol{\xi}_k) = \prod_{i=1}^D q_i^{\boldsymbol{\xi}_{k,i}}(1 - q_j)^{1-\boldsymbol{\xi}_{k,i}}$ where $q_i$ is the prior probability for the $i$-th dimension to be active. As we represent the probability distribution $\hat{p}^0(\boldsymbol{\xi})$ by a point cloud, any prior distribution can be used to insert prior knowledge. We use the same SMC sampler as Cuturi et al. (2020).

**Batch evaluations.** Several distinct groups can often be selected with close to optimal mutual information by running the forward-backward algorithm again, excluding already selected groups. When possible, we evaluate several groups at once, reducing how often we need to perform the resampling procedure and allowing the user to run several black-box evaluations in parallel. We continue adding groups to evaluate until we have reached a user-specified upper limit or until the MI of new groups is sufficiently much smaller than for the best group.

**The `GTBO` algorithm.** With the individual parts defined, we present the complete procedure for `GTBO`. `GTBO` iteratively selects and evaluates groups for $T$ iterations or until convergence. We consider it to have converged when the posterior marginal probability for each variable $\hat{p}^t(\xi_i)$ lies in $[0, C_{\text{lower}}] \cup [C_{\text{upper}}, 1]$, for some convergence thresholds $C_{\text{lower}}$ and $C_{\text{upper}}$. More details on the group testing phase can be found in Algorithm 1 in Appendix A.

Subsequently, their marginal posterior distribution decides which variables are selected to be active. A variable $i$ is considered active if its marginal is larger than some threshold, $\hat{p}^t(\xi_i) \geq \eta$. Once we have deduced which variables are active, we perform BO using the remaining sample budget. To strongly focus on the active subspace, we use short lengthscale priors for the active variables and long lengthscale priors for the inactive variables. We use a Gaussian process (GP) with a Matérn$-5/2$ kernel as the surrogate model and `qLogNoisyExpectedImprovement` (qLogNEI, Balandat et al. (2020)) as the acquisition function. The BO phase is initialized with data sampled during the feature selection phase. Several points are sampled throughout the group testing phase that only differ marginally in the active subspace. Such duplicates are removed to facilitate the fitting of the GP.

## 4 COMPUTATIONAL EXPERIMENTS

In this section, we showcase the performance of the proposed methodology, both for finding the relevant dimensions and for the subsequent optimization. We compare state-of-the-art frameworks for high-dimensional BO on several synthetic and real-life benchmarks. `GTBO` outperforms previous approaches on the tested real-world and synthetic benchmarks. In Section 4.2, we study the sensitivity of `GTBO` to external traits of the optimization problem, such as noise-to-signal ratio and the number of active dimensions. The efficiency of the group testing phase is tested against other feature analysis algorithms in Appendix B and the `GTBO` wallclock times are presented in Table 1 in Appendix E. The code for `GTBO` is available at `https://github.com/gtboauthors/gtbo`.

### 4.1 EXPERIMENTAL SETUP

We test `GTBO` on four synthetic benchmark functions, `Branin2`, `Levy` in 4 dimensions, `Hartmann6`, and `Griewank` in 8 dimensions, which we extend with inactive "dummy" dimensions (Wang et al., 2016; Eriksson & Jankowiak, 2021; Papenmeier et al., 2022) as well as two real-world benchmarks: the 124D soft-constraint version of the `Mopta08` benchmark (Eriksson & Jankowiak, 2021), and the 180D `LassoDNA` benchmark (Šehić et al., 2022). We add significant observation noise for the synthetic benchmarks, but the inactive dimensions are truly inactive. In contrast, the real-world benchmarks do not exhibit observation noise, but all dimensions have at least a marginal impact on the objective function. Note that the noisy synthetic benchmarks are considerably more challenging for `GTBO` than their noiseless counterparts.

Since the search space center is a decent solution for `LassoDNA`, `GTBO` chooses a default configuration used in the group testing phase for each repetition uniformly at random. To not give `BAxUS` a similar advantage, we subtract a random offset from the search space bounds, which we add again before evaluating the function. This ensures that `BAxUS` cannot always represent the near-optimal origin.

To evaluate the BO performance, we benchmark against `TuRBO` (Eriksson et al., 2019) with one and five trust regions, `SAASBO` (Eriksson & Jankowiak, 2021), `CMA-ES` (Hansen & Ostermeier, 1996), `HeSBO` (Nayebi et al., 2019), and `BAxUS` (Papenmeier et al., 2022) using the implementations provided by the respective authors with their settings, unless stated otherwise. We compare against random search, i.e., we choose points in the search space $\mathcal{X}$ uniformly at random.

For `CMA-ES`, we use the `pycma` implementation (Hansen et al., 2022). For `Alebo`, we use the `Ax` implementation (Bakshy et al., 2018). To show the effect of different choices of the target dimensionality $d$, we run `Alebo` with $d = 10$ and $d = 20$. We observed that `Alebo` and `SAASBO` are constrained by their high runtime and memory consumption. The available hardware allowed up to 100 evaluations for `SAASBO` and 300 evaluations for `Alebo` for each run. Larger sampling budgets or higher target dimensions for `Alebo` resulted in out-of-memory errors. We note that limited scalability was expected for these two methods, whereas the other methods scaled to considerably larger budgets, as required for scalable BO. We initialize each optimizer with ten initial samples and `BAxUS` with $b = 3$ and $m_D = 1000$ and run ten repeated trials. Plots show the mean logarithmic regret for synthetic benchmarks and the mean function value for real-world benchmarks. The shaded regions indicate one standard error.

Unless stated otherwise, we run `GTBO` with 10 000 particles for the SMC sampler, the prior probability of being active, $q = 0.05$, and 3 initial groups for the forward-backward algorithm. When estimating the function signal and noise variance, we set the assumed maximum number of active dimensions, `max_act`, to $\sqrt{D}$. The threshold to be considered active after the group testing phase, $\eta$, is set to 0.5, and the lower and upper convergence thresholds, $C_{\text{lower}}$ and $C_{\text{upper}}$, are $5 \times 10^3$ and 0.9, respectively. We run all experiments with a log-normal $\mathcal{LN}(10, 1)$ length scale prior to the inactive dimensions[1].

We use a $\mathcal{LN}(0, 1)$ prior for the active variables, resulting in significantly shorter length scales. In the group testing phase, we use batch evaluation with a maximum of 5 groups in each batch and a maximum MI drop of 1%. Note that we still count the number of evaluations, not the number of batches, towards the budget. An analysis of the impact of some of the core hyperparameters is presented in Appendix D.

The group testing phase on 2x Intel Xeon Gold 6130 machines, using two cores. The subsequent BO phase is run on a single Nvidia A40 graphics card supported by Icelake CPUs.

### 4.2 PERFORMANCE OF THE GROUP TESTING

Before studying `GTBO`'s overall optimization performance in high-dimensional settings, we analyze the performance of the group testing procedure. In Figure 2, we show the evolution of the average marginal probability of being active over the iterations for the different dimensions. The true active dimensions are plotted in green, and the inactive ones in blue squares. For all the problems, `GTBO`

---

[1]Due to constrained computational resources, we could not yet run the synthetic experiments with a $\mathcal{LN}(10, 1)$ prior. We will update the results as soon as possible. Note that making inactive dimensions less relevant, by increasing the prior of inactive dimension, is strictly beneficial in `GTBO` on synthetic benchmarks.

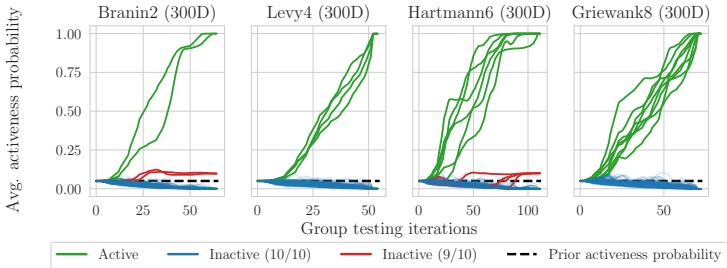

Figure 2: Evolution of the average marginal probability of being active across ten repetitions. Each line represents one dimension, and active dimensions are colored in green and inactive dimensions in blue. In the few cases where GTBO finds some inactive variables to be active, the lines are emphasized in red. The last iteration marks the end of the *longest* group testing phase across all runs. All active dimensions are identified in all runs. 6 out of 1180 inactive dimensions are incorrectly classified as active *once* in ten runs across the benchmarks, implying a false positive rate of slightly above 0.05%.

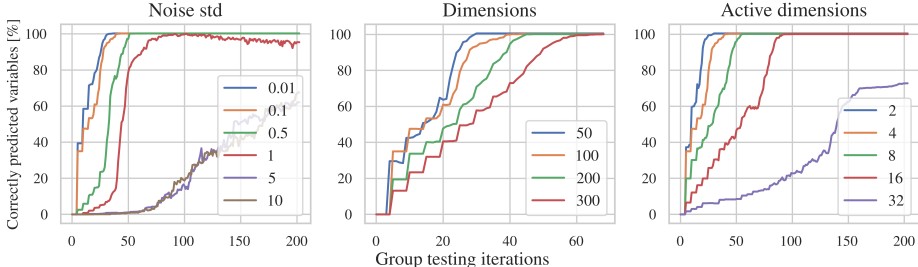

Figure 3: Sensitivity analysis for GTBO. The average percentage of correctly classified variables is displayed for increasing group testing iterations. The percentage is ablated for (left) various levels of *output noise*, (middle) number of *total dimensions*, and (right) number of *effective dimensions*. Each legend shows the configurations of the respective parameter.

correctly classifies all active dimensions during all runs within 39-112 iterations. Across ten runs, GTBO misclassifies 6 out of 1180 inactive variables to be active once each, for a false positive rate of 0.05%. The change in the number of active variables over the iterations is shown in Appendix C.

**Sensitivity analysis** We explore the sensitivity of GTBO to the output noise and problem size by evaluating it on the Levy4 synthetic benchmark extended to 100 dimensions, with a noise standard deviation of 0.1, and varying the properties of interest. In Figure 3, we show how the percentage of correctly predicted variables evolves with the number of tests $t$ for different functional properties. Correctly classified is defined here as having a probability of less than 1% if inactive or above 90% if active. GTBO shows to be robust to lower noise levels but breaks down when the noise grows too large. As expected, higher function dimensionality and number of active dimensions increases the time until convergence. Note that the signal and noise variance estimates build on the assumption that there are a maximum of $\sqrt{D}$ active dimensions, which does not hold with 32 active dimensions.

### 4.3 OPTIMIZATION OF REAL-WORLD AND SYNTHETIC BENCHMARKS

We show that identifying the relevant variables can drastically improve optimization performance. Figure 5 shows the performance of GTBO and competitors on the real-world benchmarks, Figure 4 on the synthetic benchmarks. The results show the incumbent function value for each method, averaged over ten repeated trials. We plot the true average incumbent function values on the noisy benchmarks without observation noise.

Note that Griewank has its optimum in the center of the search space. To not gain an unfair advantage, we run GTBO with a non-standard default away from the optimum. However, having the

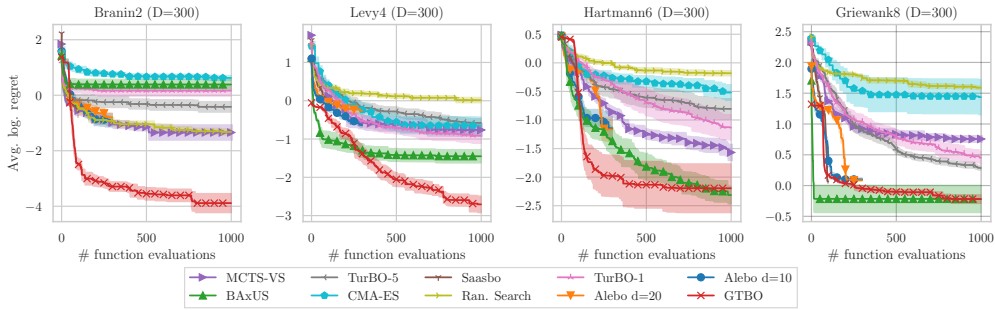

Figure 4: `GTBO` finds active dimensions and optimizes efficiently on synthetic noisy benchmarks (`Branin2`, `Levy4`, `Hartmann6`, and `Griewank8`).

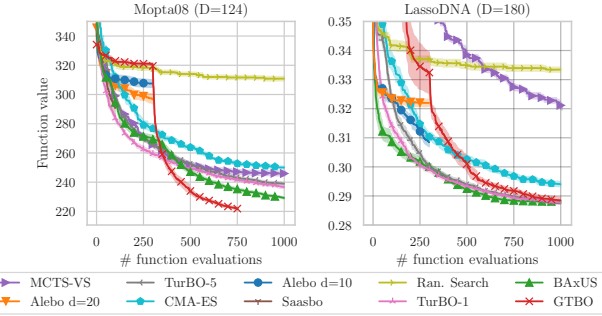

Figure 5: `GTBO` outperforms competitors in real-world experiments. Notably, the performance on `Mopta08` increases significantly right after the group testing phase at iteration 300, suggesting that the dimensions found during the GT are highly relevant. As the default configuration on `LassoDNA` performs well, the performance increase after group testing is not as visually apparent.

optimum in the center means that all possible projections will contain the optimum, which boosts the projection-based methods `Alebo` and `BAxUS`.

Figure 5 shows that `GTBO` outperforms or is on par with the state-of-the-art methods `TuRBO` and `BAxUS`. `GTBO` first identifies relevant variables, followed by a sharp drop when the optimization phase starts, indicating that knowing the active dimensions drastically speeds up optimization. In real-world experiments, the inactive variables still affect the function value. Placing long length scale priors on them allows `GTBO` to consider them for optimization if there is sufficient evidence.

## 5 DISCUSSION

Optimizing expensive-to-evaluate high-dimensional black-box functions is a challenge for applications in industry and academia. We propose `GTBO`, a novel BO method that explicitly exploits the structure of a sparse axis-aligned subspace to reduce the complexity of an application in high dimensions. `GTBO` is inspired by the field of group testing in which one aims to find infected individuals by conducting pooled tests but adapts it to the field of Bayesian optimization. `GTBO` quickly detects active and inactive variables and shows robust optimization performance in synthetic and real-world settings. Furthermore, an important by-product is that users learn what dimensions of their applications are relevant and, consequently, learn something fundamental about their application. Since `GTBO` allows for user priors on the activeness of dimensions, we will explore the potential for increasing sample efficiency by including application-specific beliefs.

**Limitations.** `GTBO` relies on the assumption that an application has several irrelevant parameters. If this assumption is unmet, the method might perform poorly or waste a fraction of the evaluation budget to identify all variables as relevant. Additionally, `GTBO` cannot exploit problems with a low-dimensional subspace that is not axis-aligned.

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

# A  THE GTBO ALGORITHM

This section describes the group testing phase of the GTBO algorithm in additional detail.

---

**Algorithm 1** Group testing phase

---

**Input:** black-box function $f : \mathcal{X} \to \mathbb{R}$, number of default configuration evaluations $n_{\text{def}}$, number of group tests $T$, number of particles $M$, prior distribution $p^0(\boldsymbol{\xi})$
**Output:** estimate of active dimensions $\boldsymbol{\gamma}$, posterior distribution $\hat{p}^T$
   **for** $j \in \{1 \ldots n_{\text{def}}\}$ **do**
      $y_{\text{def}}^{(i)} = y(\boldsymbol{x}_{\text{def}})$
   **end for**
   $\hat{f}(\boldsymbol{x}_{\text{def}}) \leftarrow \frac{1}{n_{\text{def}}} \sum_{i=1}^{n_{\text{def}}} y_{\text{def}}^{(i)}$

   split the dimensions into $3\lfloor\sqrt{D}\rfloor$ bins $B$
   **for** $j \in \{1 \ldots |B|\}$ **do**
      $y_{\text{bin}}^{(j)} = y(\boldsymbol{x}_{\text{def}} \oplus \alpha b_j)$                               ▷ a random perturbation along the dimensions in bin $b_j$
   **end for**
   sort (asc.) $y_{\text{bin}}^{(j)}$
   $\hat{\sigma}_n^2 \leftarrow \texttt{var}(y_{\text{bin}}^{(1)}, \ldots, y_{\text{bin}}^{(2\lfloor\sqrt{D}\rfloor)})$
   $\hat{\sigma}^2 \leftarrow \texttt{var}(y_{\text{bin}}^{(2\lfloor\sqrt{D}\rfloor+1)}, \ldots, y_{\text{bin}}^{(3\lfloor\sqrt{D}\rfloor)})$

   $\boldsymbol{\xi}_1, \ldots, \boldsymbol{\xi}_M \sim \hat{p}^0(\boldsymbol{\xi})$
   $\omega_1, \ldots, \omega_M \leftarrow \frac{1}{M}$                                       ▷ initial particle weights
   **for all** $t \in \{1, \ldots, T\}$ **do**
      $\boldsymbol{g}^* \leftarrow \texttt{maximize\_mi}(\boldsymbol{\xi}_1, \ldots, \boldsymbol{\xi}_M)$             ▷ find a group that maximizes MI
      $\boldsymbol{x}_t \leftarrow$ create using Eq. (1) and $\boldsymbol{g}^*$
      $z_t \leftarrow f(\boldsymbol{x}_t) + \varepsilon - \hat{f}(\boldsymbol{x}_{\text{def}})$
      $(\boldsymbol{\xi}_i, \omega_i)_{i\in[M]} \leftarrow \texttt{resample}(\boldsymbol{z}_t, (\boldsymbol{\xi}_i, \omega_i)_{i\in[M]})$
      $\hat{p}^t \leftarrow \texttt{marginal}((\boldsymbol{\xi}_i, \omega_i)_{i\in[M]})$                    ▷ get marginals
   **end for**
   $\boldsymbol{\gamma} \leftarrow (\delta_{p^T(\boldsymbol{\xi}_1) \geq \eta}, \ldots, \delta_{p^T(\boldsymbol{\xi}_D) \geq \eta})$        ▷ check which dimensions are active

---

# B  COMPARISON WITH FEATURE IMPORTANCE ALGORITHMS

We compare the performance of the group testing phase with the established feature importance analysis methods XGBoost (Chen & Guestrin, 2016) and fAnova (Hutter et al., 2014). Since fAnova's stability degrades with increasing dimensionality, we run these methods on the 100-dimensional version of the synthetic benchmarks: Branin2 (noise std 0.5), Griewank8 (noise std 0.5), Levy4 (noise std 0.1), and Hartmann6 (noise std 0.01).

Figure 6a shows the results of fAnova and XGBoost on the 100-dimensional version of Hartmann6 with added output noise. Per our results, both methods flag the third dimension as not more important than the added input dimensions (dimensions 7-100 with no impact on the function value). Additionally, fAnova seems to "switch off" the second dimension.

On Branin2 (Figure 6b), both methods detect the correct dimensions (the first and second dimensions). Furthermore, all other dimensions have zero importance, and the methods find the correct partitioning earlier than for Hartmann6. Similarly to Hartmann6, both methods fail to detect an active dimension (the fourth dimension).

On Griewank8, fAnova does not terminate gracefully. Therefore, we only discuss XGBoost for Griewank8. After 300 iterations, XGBoost only detects six dimensions reliably as active. The other two dimensions are determined to be not more important than the added dimensions. The marginals found by GTBO are shown in Figure 7. Compared conventional feature importance analysis methods, GTBO detects all active dimensions with high probability.

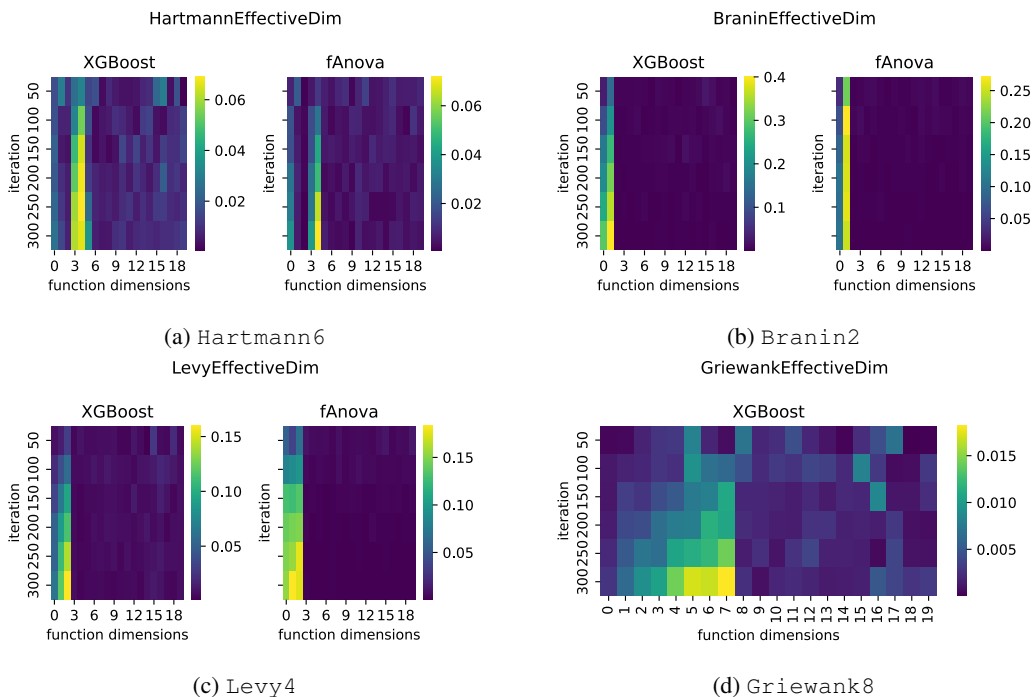

Figure 6: `XGBoost` and `fAnova` feature importance analysis on the 100-dimensional version of different synthetic benchmarks, averaged over 20 repetitions. Only the first 20 dimensions are shown.

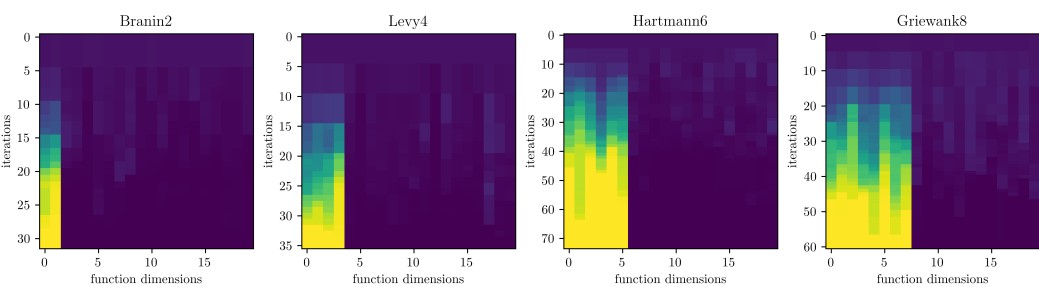

Figure 7: `GTBO`-marginals of the first 20 dimensions., averaged over 10 repetitions. If the group testing phase ends early, the last marginals are repeated to match the length of the longest group testing phase.

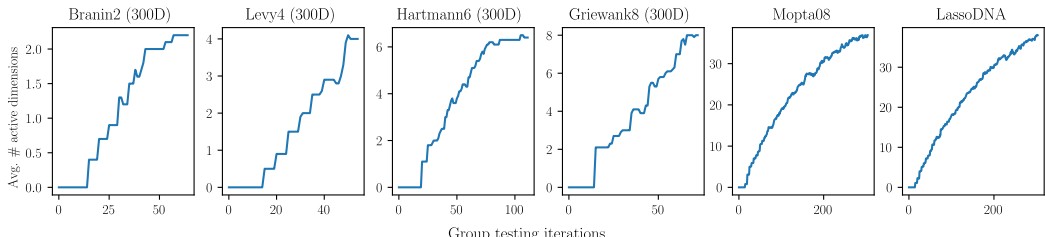

Figure 8: Evolution average number of active variables during the group testing phase (10 repetitions). The synthetic benchmarks find the correct number of active variables, whereas the real-world benchmarks find a significantly higher number.

## C  NUMBER OF ACTIVE VARIABLES

Here, we show the average number of active dimensions throughout the group testing phase. Given that the acceptance threshold of 0.5 is much higher than the initial probability of acceptance of 0.05, dimensions once considered active are rarely later considered inactive again, resulting in a close to monotonically increasing number of active dimensions.

## D  HYPERPARAMETER ANALYSIS

To measure the impact of different hyperparameters in GTBO, we run an analysis on Griewank8 in 100 dimensions. In Figure 9, we focus on the maximum batch size, initial probability of being active, the assumed number of active dimensions for estimating the signal and noise ratio, and the number of particles in the SMC sampler. The iteration at which all 20 repetitions have converged for a single setting is marked with a circle. Increasing the maximum batch size does not greatly impact the group testing performance. The initial probability is important, but both 0.05 and 0.10 perform well; the very small or large values perform worse. The assumed number of active dimensions, set to $\sqrt{D}$ in the paper, is highly important because it must be larger than the actual number of active dimensions. Otherwise, active dimensions will be used to estimate the noise, severely hampering the performance. We see this for the performance of the assumed active dimension of five, which is less than the true active dimensionality 8. Lastly, the number of particles in the SMC sampler is irrelevant for this benchmark, but it becomes more important for harder benchmarks.

We also study the impact of the parameters of the lengthscale LogNormal priors for inactive dimensions in Figure 10. We see that the performance increases for longer lengthscales. For the synthetic benchmarks, the inactive dimensions have absolutely no impact on the objective function, and as such, it is expected that de-emphasizing the importance of those variables would be beneficial.

## E  RUN TIMES

In this section, we show the average run of `GTBO`. Note that the SMC resampling is a part of the group testing phase, and as such, the total algorithm time is the GT time plus the BO time. We do batch evaluations in the GT phase described in Section 3, with a maximum of five groups tested before resampling. This significantly reduces the SMC resample time.

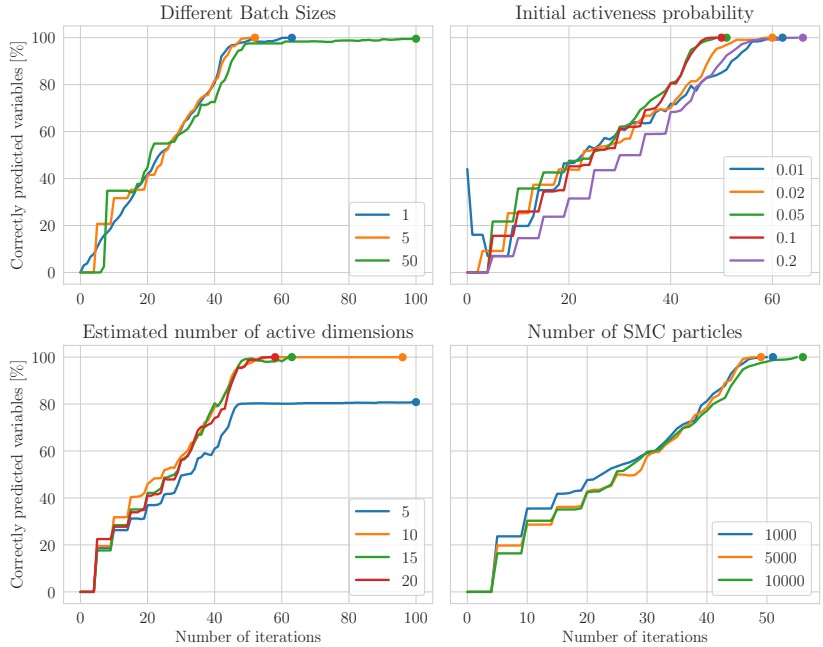

Figure 9: The percentage of correctly classified dimensions after increasing the number of iterations for different hyperparameter settings for Griewank8 (100D).

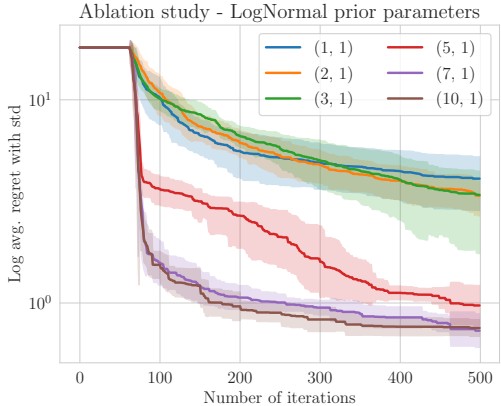

Figure 10: The log average regret of GTBO for different values of the LogNormal prior for Griewank8 (100D).

| Benchmark | GT time [h] | SMC resample time [h] | BO time [h] |
|---|---|---|---|
| `Branin2` (300D) | 1.94 | 0.583 | 9.31 |
| `Levy4` (300D) | 2.33 | 0.603 | 10.8 |
| `Hartmann6` (300D) | 3.29 | 1.14 | 14.1 |
| `Griewank8` (300D) | 2.41 | 0.846 | 9.08 |
| `Mopta08` (124D) | 5.70 | 4.74 | 7.95 |
| `LassoDNA` (180D) | 8.92 | 7.06 | 10.2 |

Table 1: Average `GTBO` runtimes. Group testing time is on the same order of magnitude as the time allocated towards BO. For BO, the $\mathcal{O}(D^2)$ complexity of Quasi-Newton-based acquisition function optimization dominates the runtime.

