# OpenReview forum: "High-dimensional Bayesian Optimization with Group Testing"
_ICLR.cc/2024/Conference — Submitted to ICLR 2024_

### Official Review · Reviewer_JQFf · 2023-10-28

**Soundness:** 3 good
**Presentation:** 3 good
**Contribution:** 2 fair
**Rating:** 6
**Confidence:** 3

**Summary:**

This submission tackles the case of high-dimensional optimization using Bayesian Optimization (BO), a sample-efficient method that has shown great success for hyperparameter tuning of large deep learning models, or in more concrete applications such as recommender systems tuning. It is therefore relevant for a venue such as ICLR, in my opinion. More particularly, the authors propose to tailor the so-called *group testing* theory to the BO setting. This is achieved by 1) extending group testing to real-valued functions and 2) dividing a usual BO run into two steps, a first step of relevant variable identification through group testing, and a second step of conventional BO, with the learned variable relevance being encoded in the Gaussian process surrogate lengthscales. The proposed method, *GTBO*, not only achieves state-of-the-art results of synthetic and real-world experiments but also enlights the practitioner with a ranking of the relevant variables, thus enhancing its understanding of the problem.

**Strengths:**

- The paper is well-written and organized in an easy-to-follow manner.
- The approach is simple and works remarkably well.
- The benchmarks include many competing methods, although some recent ones could have been considered as well, e.g. [1,2]


[1] Sparse Bayesian optimization. AISTATS 2023.
[2] Are Random Decompositions all we need in High Dimensional Bayesian Optimisation? ICML 2023.

**Weaknesses:**

- As often, the approach involves several hyperparameters, to determine whether a variable is deemed as relevant or not, and then incorporating this information in the BO statistical surrogate using carefully-designed priors.

Other than that, I have to say I cannot really spot any weakness here. I am not very familiar with the group testing framework.

**Questions:**

I am genuinely surprised by how good *GTBO* is compared to other competitors, given that the proposed approach feels suboptimal. It performs in a sequential manner, where one first does not care about function maximization, only about variable relevance, even though finding out this information is costly, and then classical BO is performed.
As the group testing phase does not care about high function values, the design evaluated in this process can be associated with low function values. When this happens, the budget has been spent on a design that does not yield a high function value, and variable relevance is learned on a part of the space we do not really care about, as it does not yield a high function value.
Any insights as to why *GTBO* seems to work despite that? Perhaps the initial starting points provided by group testing provide an accurate approximation of the function. A more relevant one than that usually obtained by uniform/Sobol sampling?

Small typo:

I would write the r.h.s. of Eq. 4 as $H(Z_t) - H(Z_t|\boldsymbol{\xi}) = H(Z_t) - \sum_{\xi \in \{0,1\}^D} p(\xi)H(Z_t|\boldsymbol{\xi}=\xi)$ instead of $H(Z_t) - H(Z_t|\boldsymbol{\xi}) = H(Z_t) - \sum_{\boldsymbol{\xi} \in \{0,1\}^D} p(\xi)H(Z_t|\boldsymbol{\xi})$ (as is done in [1]).

[1] Noisy Adaptive Group Testing using Bayesian Sequential Experimental Design, arXiv.

---

> ### Author Response · Authors · 2023-11-21
> **Reply to Reviewer JQFf**
>
> We thank the reviewer for their helpful feedback and are happy they appreciate the empirical performance of GTBO. In the following, we address the reviewer's concerns.
>
> We have happily added [1] to our literature review. However, please notice that we already discussed [2] in our related work section. We decided not to compare against it as it has shown subpar performance in another recent comparison [3] and achieved worse performance on Lasso-DNA than many competing approaches in the paper.
>
> **Hyperparameters:**
> To better understand the impact of the core hyperparameters used in GTBO, we have added an ablation analysis in Appendix D in the revised Manuscript.
>
> **Answers to Questions:**
>
> _“I am genuinely surprised by how good GTBO is compared to other competitors, given that the proposed approach feels suboptimal.”_
>
> Our approach is based on one fundamental premise: If the effective subspace assumption holds true, then spending time finding the correct assignment pays off in the subsequent Bayesian optimization phase. Whether the assumption itself is true is debatable, but we believe GTBO goes a long way in assessing its sensibility.
>
> _“When this happens, the budget has been spent on a design that does not yield a high function value, and variable relevance is learned on a part of the space we do not really care about, as it does not yield a high function value.  Any insights as to why GTBO seems to work despite that?”_
>
> The reviewer’s question addresses another relevant assumption regarding stationarity (i.e., assuming the GP's hyperparameters to be spatially invariant).
>
> In high-dimensional optimization, we believe assessing whether a dimension is relevant is as good as one can reasonably do within a sensible budget (and even that may be challenging). We ultimately believe that the assumption that a variable that is active somewhere is also likely to be active everywhere is sensible and that it suffices that GTBO effectively identifies active variables locally, whereas other approaches may fail to do so at all. Once identified, GTBO can quickly identify high-value regions and better fine-tune the hyperparameters of the active dimensions once a sufficient amount of high-value points are found.
>
> [1] Sparse Bayesian optimization, Liu et al., AISTATS 2023. \
> [2] Are Random Decompositions all we need in High Dimensional Bayesian Optimisation? Ziomek et al., ICML 2023. \
> [3] Bounce: Reliable High-Dimensional Bayesian Optimization in Combinatorial and Mixed Spaces, Papenmeier et al., NeurIPS 2023

---

### Official Review · Reviewer_fy4F · 2023-10-30

**Soundness:** 2 fair
**Presentation:** 3 good
**Contribution:** 2 fair
**Rating:** 5
**Confidence:** 3

**Summary:**

This paper proposes GTBO, a high-dimensional Bayesian optimization method which first uses adaptive group testing to identify active dimensions and then optimize over the active variables. The authors extend the binary group testing method to continuous space via maximizing the multual information estimated from Gaussian process. Experiment on synthetic functions and two real-world benchmarks demonstrate the efficiency of GTBO.

**Strengths:**

1. The proposed group testing idea is clear and easy to follow.

2. The experiment shows that using group testing can efficiently identify active variables.

**Weaknesses:**

1. I think the experiment comparison is not fair, where the best initial point of GTBO is always better than baselines, which gives additional advantage to GTBO.

2. I think the work of MCTS-VS[1] is also a HDBO method using variable selection, which is a similar work as GTBO and should be added into the baselines.

[1] Song, Lei, et al. "Monte carlo tree search based variable selection for high dimensional bayesian optimization." Advances in Neural Information Processing Systems 35 (2022): 28488-28501.

**Questions:**

1. As mentioned in weakness part, the initial points are different between GTBO and other baselines. Can you show the result when using same initial points in baselines?

2. Why choosing the search center as the default configuration? What is the performance of GTBO when choosing default configuration as other position (e.g. best point in the random sampled initial dataset)?

3. What is the search bound in the benchmark you used? Does GTBO utilize the advantage of symmetric search space as BAxUS?

4. What is the batch size used in the experiment? The paper mention that "GTBO integrates well with batch BO pipelines with little to no performance degradation". Is there any experiment result to support this statement?

---

> ### Author Response · Authors · 2023-11-21
> **Reply to Reviewer fy4F**
>
> We thank the reviewer for their feedback and are happy that the reviewer appreciates the simplicity of our method. We will now address the reviewer's comments and concerns.
>
> **Q1, Q2. Unfair advantage due to default configuration**:
> We used the center of the search space as the default location because this is often considered the most natural choice in BO, and GTBO requires a default configuration. We hope that the reviewer agrees on this point. It was not intentionally chosen to create an unfair advantage.
>
> For the synthetic benchmarks, the initial points here are mostly irrelevant — we see that all methods reach the level of the GTBO default configuration within 5-10 iterations of random sampling. This gives GTBO, at best, ten iterations a head start. With a sample budget of 1000 evaluations, that has negligible impact. The same holds for Mopta08. Furthermore, the optimum is well-known for all synthetic benchmarks, and we ensured that it never coincides with our default configuration.
>
> However, we acknowledge that for Griewank and LassoDNA, using the search space's center could imply an unfair advantage.
> In Griewank, the optimum lies in the center of the search space. As such, we intentionally altered the default configuration used in GTBO.
> For LassoDNA, we have run new experiments with random default locations and added them in Figure 5 of the newly uploaded version of the manuscript. BAxUS also benefits from a near-optimal solution in the center because it can always represent that point, regardless of the embedding size. We, therefore, also changed the search space bounds for BAxUS by a random offset, for which we account when evaluating the function. GTBO is on par with the other methods, even though it spends a large fraction of the evaluation budget on the group testing phase.
>
> **MCTS-VS:**
> MCTS-VS is already present in Figures 3 and 4 (in the original submission) and can be considered moderately competitive.
>
> **Q3. Search bound:** We use the standard search space defined by the benchmark problems normalized to the unit hypercube. If you are referring to BAxUS & HeSBO potentially benefiting from the optimum being identically located across dimensions (i.e., in 1, 1, 1, 1… for Levy), GTBO will not benefit from such properties of the objective, as dimensions are not coupled together.
>
> **Q4. Batch size:** We have improved the description of how batching is performed in the GT phase in the revised manuscript, as well as added an ablation analysis on its effects in Appendix D. We currently do not use any batch methods for the BO part, but several approaches could be readily integrated into the BO phase such as for example [1].
>
> [1] González, Javier, et al. "Batch Bayesian optimization via local penalization." Artificial intelligence and statistics. PMLR, 2016.

---

> ### Comment · Reviewer_fy4F · 2023-11-22
>
> Thanks for your reply , which clarifies part of my concerns. I still have the following questions:
>
> **"We see that all methods reach the level of the GTBO default configuration within 5-10 iterations of random sampling."**
>
> I am hesitant to agree this point. In Levy 4, Griewank8 and LassoDNA, seems that the random search is never able to approach the initial level of GTBO.  Some baselines also take more than 10 iterations to achieve this level (e.g. all the baselines except BAxUS in LassoDNA in the original Figure 5). I encourage the authors to set same level of initial point when comparing against the baselines.
>
> I also observe that in the new Figure 5, the performance of GTBO degrades badly after changing the search space bounds to deviate from the optimal region. Seems that the performance of GTBO also sensitive to whether the default configuration lies near the optimum region. I encourage the authors to conduct some ablation studies about this point.
>
> **"We use the standard search space defined by the benchmark problems normalized to the unit hypercube."**
>
> Can you elaborate the standard unnormalized search bound and the optimal position in the benchmarks you used?
>
> **Batch Size**
>
> In the new Figure 9, the performance is shown respect to the number of iterations. Does this mean the performance degrades in terms of the evaluation number?

---

> ### Author Response · Authors · 2023-11-22
> **Reply to Comment**
>
> **Good initial performance on Levy and Griewank.** \
> This is an interesting point. Note that regardless of the initial solution, GTBO shows a sharp increase in performance after the group testing phase. Overall, GTBO is the only method showing this behavior. We cannot rule out that GTBO has a headstart on certain benchmarks, but this is not intentional, and we already changed the default configuration for benchmarks where we know that the search space center is a good solution. For Branin and Hartmann, where the initial solution is poor and below the performance of random search, GTBO still eventually outperforms or is on par with the baselines. We agree, however, with the reviewer that this point may cause unnecessary suspicion of an unfair advantage. We will address this in the camera-ready version by running an additional experiment where we run all benchmarks with random default configurations.
>
> **Performance of Lasso** \
> It is true that the updated results for LassoDNA, while still state-of-the-art, are slightly worse than the previous iteration. We thank the reviewer for pointing this out and are happy to present a fairer picture. After the group testing phase, GTBO needs significantly fewer function evaluations to reach the same level of performance as BAxUS.
>
>
> **Benchmark bounds** \
> Each benchmark has a predefined search space that we normalize to the unit cube $[0, 1]^d$. In the following, we write the unnormalized bound and the corresponding unnormalized default and optimal solutions.
>
> **Branin:**
> Bounds: $[[-5,10], [0, 15]]$ \
> Default: $(2.5, 7.5)$ \
> Optima: $(-\pi, 12.275)$, $(\pi, 2.275)$, and $(9.42478, 2.475)$.
>
> **Levy:**
> Bounds: $[-10,10]^d$ \
> Default: $(0,0,\ldots,0)$ \
> Optimum: $(1,1,\ldots,1)$
>
> **Hartmann:**
> Bounds: $[0,1]^6$ \
> Default: $(\frac{1}{2},\frac{1}{2},\ldots,\frac{1}{2})$ \
> Optimum: $(0.20169, 0.150011, 0.476874, 0.275332, 0.311652, 0.6573)$
>
> **Griewank:**
> Bounds: $[-600, 600]^d$ \
> Default: $(100,100,\ldots,100)$ \
> Optimum: $(0,0,\ldots,0)$
>
> **Mopta**:
> Bounds:  $[0,1]^{124}$ \
> Default: $(\frac{1}{2},\frac{1}{2},\ldots,\frac{1}{2}.5, .5,..)$
>
> **LassoDNA**:
> Bounds: $[0,1]^{180}$ \
> Default: Uniformly random in the full search space.
>
>
>
> Regarding question 9
> The figure shows the percentage of dimensions converging to their correct value (true/false, y-axis) per iteration (x-axis). Initially, no dimensions have converged as they all start with a prior probability of being active of 0.05. After an increasing number of iterations, more and more are successfully classified. We aim to classify all correct in as few evaluations as possible, as that lets us allocate more evaluations to the BO phase.
>
> Consider the top-left subfigure. Here, the lines show the behavior for different maximum batch sizes. The line ends (marked with a circle) when all 20 repetitions are completed, i.e., all marginals have converged. The convergence rate is nearly identical for the different batch options. However, when using a maximum batch size of 50 (green line), one out of the 20 repetitions did not converge within 100 evaluations, which explains the long tail. If not for that one run, the line representing batch size 50 would have ended after the same number of iterations as the yellow. We will describe this in the revised manuscript. Hence, the batching seems to have a negligible detrimental impact on the performance.

---

> > ### Comment · Reviewer_fy4F · 2023-11-22
> >
> > Thanks for your reply. I understand that the different initial values are not intentional. But I still consider the default configuration is an important component in GTBO. Given the results after randomizing the default configuration, I encourage the authors to conduct further ablation study on the choice of the default configuration.
> >
> > Overall, I've increased my score from 3 to 5.

---

### Official Review · Reviewer_yfZG · 2023-10-30

**Soundness:** 3 good
**Presentation:** 3 good
**Contribution:** 3 good
**Rating:** 5
**Confidence:** 3

**Summary:**

This paper introduces an algorithm for high-dimensional Bayesian optimization (BO), called GTBO (Group Testing Bayesian Optimization). The algorithm explicitly divides high-dimensional BO into two steps: in the first, a set of group testing experiments are run to probabilistically identify inactive input dimensions. In the second, BO is run with a relatively standard method (Matern 5/2 with qLogNEI acquisition function), while applying different length-scale priors for active vs. inactive dimensions.

------ AFTER AUTHOR RESPONSE -----

Thanks for these clarifications. I definitely misunderstood how the GT iterations were treated in the computational results, which the authors have pointed out. I have raised my Contribution and Overall scores to account for this mis-judging of the results.

There are still some points I believe could be improved--the authors also note both points in their response. Firstly, the presentation could focus on this as a feature selection method rather than a BO algorithm. It is a great feature that GTBO actively selects sampling points in contrast to "traditional feature selection methods," but the entire GT step is still de-coupled from the latter BO step, unless I am mistaken here. Secondly, it would be nice to see some more realistic benchmark problem(s).

**Strengths:**

1. The presentation of the group testing methodology clearly explains the mathematical foundation and practical implementation details of the proposed algorithm.
1. The computational tests are evaluated on several synthetic problems and also real-world benchmarks.

**Weaknesses:**

1. While this work is presented as a new BO algorithm, it is effectively a feature selection algorithm. The proposed group testing algorithm could select active dimensions to be optimized by any standard BO algorithm (e.g., ignoring the inactive dimensions). Likewise, a different feature selection method could be followed by the employed BO, which is relatively standard.
1. The motivation for the benchmark problems needs to be strengthened. The synthetic benchmarks have 2-8 active dimensions and approximately 300 active dimensions without justification for this setting. The real-world benchmarks have no noise, again without justification.
1. From what I understand, the comparisons for GTBO do not include the 39-112 iterations for group testing, which are used as the initial sample points for BO. Therefore, in Figs 3-4, where the other algorithms are starting from 0 function evaluations, GTBO is already starting at many.

**Questions:**

1. The description of batch evaluations on pg 6 needs significant clarification, i.e., how many is “several,” and how close is “close”? If batch sampling is available, why doesn’t the user just use the maximum number of batches for every group?
1. How are the log-normal length scales for inactive dimensions determined?

---

> ### Author Response · Authors · 2023-11-21
> **Reply to Reviewer yfZG**
>
> We are happy that the reviewer appreciates our experimental evaluation and thank them for their helpful feedback. In the following, we address the reviewer's concerns:
>
> **The GTBO method:**
>
> The reviewer is correct in that the proposed method is a feature selection method similar to MCTS-VS, HeSBO (random, axis-aligned feature selection), and even REMBO (random, non-axis-aligned feature selection method followed by vanilla BO).
>
> However, GTBO is the first method to fully exploit the axis-aligned effective dimensionality assumption by learning the effective dimensions with a principled method for identifying active variables and then optimizing the identified effective subspace. GTBO was designed so that If such an effective subspace existed in practice, GTBO would make maximal use of the assumption to solve the problem. As such, GTBO truly exposes whether the effective dimensionality assumption is a good one in a manner that SAASBO, BAxUS, and MCTS-VS only partially do. We hope that the reviewer appreciates the scientific value of this approach.
>
> Lastly, we politely disagree that feature selection followed by BO in itself is standard, as many high-dimensional BO methods adaptively try to learn the effective subspace on the fly (SAASBO, BAxUS, MCTS-VS) or rely on initial randomness (HeSBO, REMBO). Further, the traditional feature selection methods referenced in the paper use a pre-selected data set. In contrast, GTBO sequentially selects which samples to evaluate to make maximal use of the sample budget.  If the reviewer believes we are incorrect, we would greatly appreciate it if the reviewer could specify such works further.
>
> **Benchmarks:**
> We agree with the reviewer that the artificially sparse synthetic benchmarks are likely not very realistic. However, they are the de-facto synthetic benchmark suite used in modern high-dimensional Bayesian optimization studies [1, 2, 3, 4, 5].  While it seems unsurprising that GTBO performs well on these synthetic benchmarks, we emphasize that it outperforms the state-of-the-art on both real-world benchmarks. Regarding the noise, we used two commonly run high-dimensional real-world benchmarks [1, 3, 4, 5], which happened to be noiseless.
>
> **Counting GT iterations towards the total:**
> This is a misunderstanding; the group testing iterations do count towards the total. We perform the same number of evaluations as all baselines. The flat regions in the GTBO curve in Figure 4 are these group testing iterations. We made this more explicit in the revised manuscript in Section 4.3.
>
> _“GTBO first identifies relevant variables, followed by a sharp drop when the optimization phase starts, indicating that knowing the active dimensions drastically speeds up optimization.”_
>
> **Question 1 (batch evaluations):** To reduce the run time of the experiments, we used batching with up to 5 parallel evaluations. We additionally have a safeguard that only allows batching if the difference in mutual information between the best and worst groups is less than 1%. Note that we still count the number of evaluations, not the iterations, to keep the conditions equal for all methods. We thank the reviewer for pointing out the lack of description. We have updated the manuscript with more details and added a short analysis in the appendix.
>
> **Question 2 (log-normal prior for inactive dimensions):** The exact parameters of the priors were not the paper's main focus and were chosen based on initial experimentation and values found in the popular BoTorch library. In particular, the mode of our LogNormal(0,1) lengthscale prior ($\approx 0.367$) roughly matches the mode of BoTorch’s default Gamma(3,6) prior ($\frac{1}{3}$). A brief analysis of the impact of the lengthscale parameters is added in Appendix D.
>
> [1] High-Dimensional Bayesian Optimization with Sparse Axis-Aligned Subspaces, Eriksson and Jankowia, UAI 2021 \
> [2] Re-Examining Linear Embeddings for High-Dimensional Bayesian Optimization, Letham et al., NeurIPS 2020 \
> [3] Increasing the Scope as You Learn: Adaptive Bayesian Optimization in Nested Subspaces, Papenmeier et al., NeurIPS 2022 \
> [4] Self-Correcting Bayesian Optimization through Bayesian Active Learning, Hvarfner et al., NeurIPS 2023 \
> [5] Bounce: a Reliable Bayesian Optimization Algorithm for Combinatorial and Mixed Spaces, Papenmeier et al., NeurIPS 2023

---

### Official Review · Reviewer_9ChC · 2023-10-31

**Soundness:** 3 good
**Presentation:** 2 fair
**Contribution:** 3 good
**Rating:** 6
**Confidence:** 3

**Summary:**

The paper introduces GTBO which introduces ideas from feature selection literature and Group testing theory to the problem of selecting relevant features to reduce dimensionality of high dimensional BO problems and reduce the effect of pathology of curse of dimensionality for such problems. The total budget is divided into two halves, first the relevant features are identified using first set of function evaluations to create an active subspace of features, and then by assigning different priors on lengthscales for each set of relevant and irrelevant features the remaining budget is used for BO. Experiments are carried out on simulated popular datasets and two real world datasets and the performance of proposed method is compared against existing algorithms. The proposed model is probabilistic and can handle noisy observations. The method seems to do well and with the inherent advantage being that it is more interpretable than projection based dimenionality reduction methods.

**Strengths:**

1. The paper is mostly well written.
2. The baselines and relevant literature is well covered and duly introduced to the readers.
3. The results from the experiments suggest that the nmethod works well as compared to baselines both on simulated and real world datasets.
4. I think it is a strenght of the method that it combines the advantages of interpretibility which come along with feature selection compared to projection based approaches so the user gets to understand his data as he is performing BO.
5. The math and equations look fine to me.
6. It is great the authors carry out and report sensitivity analysis and ablation study in Appendix and main paper.

**Weaknesses:**

1. The model makes many assumptions that the features are relatively independent, since for highly correlated features, it might not be possible to break them into sets of active and inactive features without knowing their correlations beforehand. Assuming that the probabilities of dimensions to be active are independent, is rather a strong simplifying assumption and will not hold in many practical datasets and situations.
2. The paper does not list its own limitations properly.
3. Certain choice of parameters for instance :  $\sqrt(D)$ to be the value of active dimensions, choice of prior: logNormal with particular values of location and scale parameters can be better motivated. Why logNormal and not Gamma for instance, which is common hyperprior for lengthscale ?
4. Maybe have one more real world dataset.
5. The writing of the Experiment section can be improved, because somehow the flow of information is not good, as the authors introduce the figures in a weird order (Minor) and do a bit of back and forth.
6.. Maybe make the lines in plots thicker.

**Questions:**

Minor comments and questions:
1. What is the value of $C_{lower}$ and $C_{upper}$, apologies if I missed it.
2.  Why did the authors use the particular acquistion function which they did ?

---

> ### Author Response · Authors · 2023-11-21
> **Reply to Reviewer 9ChC**
>
> We thank the reviewer for their helpful feedback and are glad they appreciate our approach and found the paper well-written. In the following, we address the reviewer's concerns.
>
> **Assumption of independence :**
> This question is interesting, but we do not make explicit assumptions about independence in the GT or BO phases. The GT algorithm natively models the dependence of variables in the GT phase.
>
> For a brief example, assume that a variable $x_1$ only impacts the search space for very specific values of another variable $x_2$ and vice versa. In this case, our belief of whether $x_1$ is active is not independent of whether $x_2$ is active. This dependence is captured by the joint distribution, i.e., $p(\xi_1, \xi_2) \neq p(\xi_1,)p(\xi_2)$.
>
> When optimizing the mutual information, this dependence is also taken into account.
>
> In the group testing phase, we consider a probability distribution over which variables are active. As we, in several places in the manuscript, show the marginal distribution over the probability of individual variables being active, we can see that there is a risk that the reader confuses this with the fact that the joint probability distribution assumes independence. However, this is not the case - the joint probability over which variables are active does not assume any independence between variables. This joint probability distribution has no closed-form expression, so we need the sequential Monte Carlo sampler.
>
> **Limitations :**
> We thank the reviewer for the suggestion and have added an explicit limitations section to the newly uploaded manuscript. For ease of access, we report it here as well:
>
> "GTBO relies on the assumption that an application has several irrelevant parameters. If this assumption is unmet, the method might perform poorly or waste a fraction of the evaluation budget to identify all variables as relevant.
> Additionally, GTBO cannot exploit problems with a low-dimensional subspace that is not axis-aligned."
>
> **Hyperparameters:**
> We agree with the reviewer that several hyperparameters deserve more attention than they were given in the original manuscript. However, from initial testing, our experience is that GTBO is not particularly dependent on those hyperparameters. We have now added a hyperparameter analysis to the appendix of the revised manuscript.
>
> Gamma and log-normal priors have been shown to behave similarly [e.g., 1, 2], with the log-normal prior having heavier tails. We chose a log-normal prior because it has an easier interpretation, but we believe that a Gamma prior would have also served our purpose [1]. Studying the impact of different priors is not the goal of this paper, so we refrain from making that analysis. We will, however, motivate this choice more clearly in the revised manuscript.
>
> We further agree that the assumed maximum number of active variables is an important hyperparameter of our method, for which $\sqrt{D}$ has shown to be a robust estimate in our experiments. We have now added a discussion on this and an ablation study to the appendix in the revised version of the manuscript.
>
> **Additional real-world benchmarks:**
> We agree that additional benchmarks are relevant. We are working on setting up an additional real-world benchmark for the camera-ready version.
>
> **Reorder results section:**
> We improved the logical flow of the experimental section by integrating the sensitivity analysis in the subsection of the group testing phase. This way, we start with the experimental setup, discuss the effect of the group testing phase, and finally, the overall optimization performance. These changes are reflected in the revised version of the manuscript submitted with this rebuttal.
>
> **Maybe make the lines in the plots thicker:**
> We thank the reviewer for pointing this out. We have increased the thickness of the lines in Figure 2.
>
> **C_lower and C_upper:**
> $C_{\textrm{lower}}$ and $C_{\textrm{upper}}$ define are the bounds for when we consider a dimension to have converged. In other words, if the marginal probability of a dimension to be active is below $C_{\textrm{lower}}$ or above  $C_{\textrm{upper}}$, we consider it converged, and when all dimensions have converged we end the group testing phase. They are described in The GTBO algorithm in Section 3, and the values we use are written in Section 4.1.
>
> **Acquisition function:**
> qNoisyEI is one of the most popular acquisition functions used in BO today. It is an extension of the traditional Expected Improvement (EI), and performs better on noisy problems. It has an efficient implementation in the BoTorch BO library (https://botorch.org/).
>
> [1] Grain size distribution: The lognormal and the gamma distribution functions, Vaz and Fortes, 1988 \
> [2] A Comparison of Gamma and Lognormal Distributions for Characterizing Satellite
> Rain Rates from the Tropical Rainfall Measuring Mission, Cho et al., 2004

---

### Author Response · Authors · 2023-11-21
**Global comment to all reviewers**

We want to thank all reviewers for their comments and helpful feedback. This has helped us improve the paper significantly.

We have decided to change the length scale prior to the inactive dimensions to a log-normal LN(10,1) prior as this yields an even stronger performance in Figure 10 in the new ablation analysis in the revised manuscript.  We are still re-running some experiments. We, therefore, only show the updated real-world experiments. The synthetic benchmarks should not sustain any loss from increasing the prior, as it further decreases the importance the model assigns to the inactive dimensions. Note that the Mopta08 runs haven’t finished yet but are sufficient to show GTBO’s superiority. We are confident that we can show the synthetic results by the end of the discussion period, but we would like to post our replies now to give the reviewers a chance to engage.

Based on the reviewer’s feedback, we decided to change the default configuration for the LassoDNA benchmark since the search space center seems to be a disproportionally good solution. Instead, we now sample a default configuration uniformly at random for each repetition.  Since BAxUS has a similar advantage (its embedding can always represent the original in the original space, regardless of the embedding size), we decided to change the search space bounds by a random offset so that the search space becomes asymmetric around the origin. When evaluating a point, we account for that so that the overall search space bounds remain unchanged.

---

### Meta-Review · Area_Chair_jGdS · 2023-12-06

**Metareview:**

This paper explores Bayesian optimization in the sparse axis-aligned subspace setting. Overall, the reviewers placed this paper really at the borderline. In favor of the paper are pretty strong empirical results, that are honestly even surprising considering the complete lack of interaction between the feature selection phase and the Bayesian optimization phase. Ultimately, this paper is arguably even just a GP feature selection paper with a BO algorithm tacked on at the end. To Reviewer JQFf's point, I think it would be trivial to construct synthetic functions for which the authors' approach fails in practice for this reason -- simply construct a function where some dimensions are irrelevant except when very close to local or global optima. Granted, this behavior may not actually occur in practice, and the authors' ideas otherwise seem reasonable.

When reviewing the paper myself to resolve the borderline nature of the paper, I ultimately do think that the authors' method is quite interesting and worthy in its own right of publication. However, I do think that there are a few things the authors absolutely need to address:

- In both problems in Figure 5, it is clear that several methods have not converged yet. Indeed, on LassoDNA, the authors method has only just caught up to the other optimization methods. What happens next?

- The authors' results with SAASBO are significantly worse than reported in the original paper. On the MOPTA design problem, Figure 4 in the SAASBO paper reports objective values of around 230 after 300-400 evaluations. These values are apparently not even achieved by 1000 evaluations. This discrepancy is significant, as manually superimposing the plot from Figure 4 in the SAASBO paper on top of the authors' Figure 5 seems to suggest SAASBO is *at least* competitive?

- The authors mention during the author feedback period a significant number of updates to their own experimental results and experimental evaluation settings, with some results finished and some results not. Finish these updates and get the paper reviewed again. Especially in light of the previous bullet point.

**Justification For Why Not Higher Score:**

The authors method is really interesting, and it's almost surprising how well it works, but I think there are a number of experimental discrepancies that absolutely need to be addressed before the paper is ready for publication.

**Justification For Why Not Lower Score:**

N/A

---

### Decision · Program_Chairs · 2024-01-16

Reject